# Correction of Light and Heavy Hydrocarbons and Their Application in a Shale Oil Reservoir in Gaoyou Sag, Subei Basin—A Case Study from Well SX84

Qi Zhi [1], Shuangfang Lu [2,3,*], Pengfei Zhang [4], Hongsheng Huang [1], Junjie Wang [1] and Zizhi Lin [1]

1 School of Geoscience, China University of Petroleum (East China), Qingdao 266580, China
2 Sanya Offshore Oil & Gas Research Institute, Northeast Petroleum University, Sanya 572025, China
3 Key Laboratory of Continental Shale Hydrocarbon Accumulation and Efficient Development, Ministry of Education, Northeast Petroleum University, Daqing 163318, China
4 College of Earth Science and Engineering, Shandong University of Science and Technology, Qingdao 266590, China
* Correspondence: lushuangfang@nepu.edu.cn

**Abstract:** To accurately evaluate the shale oil resources in the Funing Formation of the Gaoyou Sag, Subei Basin, light and heavy hydrocarbon correction models of $S_1$ were developed based on the rock pyrolysis of liquefrozen, conventional, and oil-washed shales. The improved ΔlogR technique was applied to establish the *TOC*, $S_1$, and $S_2$ logging evaluation methods. The results showed that the $S_2$ values after oil washing were significantly lower than before. The difference between these two $S_2$ (Δ$S_2$) values is the heavy hydrocarbon correction amount of $S_1$, which is about 0.69 $S_2$. There was almost no loss of light hydrocarbons during liquefrozen shales' pyrolysis tests; the ratio of liquefrozen to conventional $S_1$ values is the light hydrocarbon correction factor, which is about 1.67. The corrected $S_1$ is about 3.2 times greater than the conventional shale-tested value. The $S_1$ and *TOC* are obviously "trichotomous"; a *TOC* greater than 1.5% and corrected $S_1$ larger than 4.0 mg/g corresponds to the enriched resource. The logging estimated results show that the total shale oil resources in the $E_1f_2$ of the Gaoyou Sag are about 572 million tons, of which the enriched resource is about 170 million tons.

**Keywords:** light and heavy hydrocarbon correction; classified resource; resource evaluation; shale oil; Gaoyou Sag

## 1. Introduction

Unconventional oil and gas have been of worldwide concern for oil and gas exploration. Based on the "thermal triangle" theory, conventional and unconventional resources in the 25 basins in North America were evaluated, showing that the recoverable resources of unconventional resources far exceeded the conventional resources [1]. Although this method is imprecise, it can be seen that unconventional oil and gas resources have excellent prospects. Meanwhile, shale oil has received more attention for unconventional energy exploration in the world.

Shale oil refers to the oil contained in a shale series dominated by shale, including the oil in shale pores and fractures, as well as the oil in the adjacent layers and interlayers of tight carbonate or clastic rocks [2,3]. Shale oil is the hydrocarbon resource formed in a mature shale series (including mudstone and shale), mainly occurring in the free and adsorbed states, but with few dissolved states. It is characterized by continuous distribution, considerable resources, and a long production cycle [4,5].

In recent years, shale oil exploration in North America has developed rapidly and commercial production of shale oil has been achieved in the United States. There have been significant shale oil discoveries in central Alberta and southern Texas, as well as other favorable areas such as the Los Angeles region, the south of the Gulf of Mexico, and eastern

Canada Shale oil development has significantly increased oil production in the United States and Canada [6]. According to the U.S. EIA, shale oil production will increase to $7.1 \times 10^6$ barrels per day by 2040 [7]. Moreover, shale oil shows huge exploitation potential in China, such as in the Ordos, Sichuan, Songliao, Subei, and Bohai Bay basins [8–11].

Resource assessment is essential in shale oil exploitation. The organic matter in shale consists of two parts: extractable organic matter (shale oil) and non-extractable organic matter (kerogen). Shale oil includes hydrocarbon and non-hydrocarbon NSO compounds. Chloroform-extractable bitumen "A" is most similar to shale oil in composition, with hydrocarbons and NSO compounds. However, most of the $C_{6–13}$ light hydrocarbons are vaporized during the volatilization of chloroform in the chloroform-extractable progress.

$S_1$ represents the fraction of the hydrocarbons in shale oil and is a standard parameter used to evaluate the shale oil content. $S_1$ refers to the volatilized hydrocarbons during the rock pyrolysis analysis when temperatures are less than 300 °C, which are $C_{7–33}$ hydrocarbons. However, shale light hydrocarbons ($C_{6–13}$) have primarily dissipated due to the long placement time [12]. $S_2$ represents the generated hydrocarbons from kerogen when the temperature gradually rises from 300 °C to 600 °C during the rock pyrolysis. However, some heavy hydrocarbons already present in the shale have boiling points much higher than 300 °C. Moreover, the adsorption and swelling of kerogen cause it to be impossible for the residual hydrocarbons to be completely volatilized before 300 °C [13], corresponding to this residual hydrocarbon being detected as $S_2$. Therefore, due to the storage conditions of the cores and experimental testing techniques, $S_1$ lost some light and heavy hydrocarbons, resulting in $S_1$ being significantly lower than the actual in situ shale oil content. Thus, the light and heavy hydrocarbon correction for $S_1$ is required to evaluate the total shale oil content.

Shales exhibit low porosity and permeability, causing shale oil development to be difficult, and not all resources can be extracted efficiently [14–17]. Under the current economic and technological conditions, only part of the shale oil resources can be effectively exploited [18]. Therefore, it is necessary to conduct a graded evaluation of shale oil resources to clarify the enriched resource distribution and guide shale oil exploration and development. Lu et al. indicated that the shale oil content ($S_1$ or "A") showed "trichotomy" with the increase in *TOC* [19]. Specifically, when the *TOC* is large, $S_1$ is characterized by relatively stable large values, corresponding to the enriched resources or saturated resources; however, if *TOC* is lower, $S_1$ locates in the low value-region, associated with the dispersed or saturated resources; if *TOC* is between these two regions, $S_1$ increases significantly and is intermediate, which is called low-efficiency or undersaturated resources. However, the current shale oil resource classification criteria are mainly for residual shale hydrocarbons and resource classification evaluation criteria based on the total shale oil content have not been established.

The logging evaluation method has the advantages of a shorter cycle, higher resolution, and more visual and continuous characteristics. There is a certain response relationship between various logging parameters and *TOC*, $S_1$, and $S_2$. Therefore, the prediction model of *TOC*, $S_1$, and $S_2$ quantitative prediction can be established. Domestic and foreign scholars have proposed a variety of methods to predict the *TOC* of hydrocarbon source rocks based on logging data, such as the multiple linear regression method, ΔLogR method, BP neural network method, etc. Different methods have different principles, advantages, and disadvantages. They are applicable to different geological conditions and data conditions.

The shale oil and gas resource evaluation methods mainly include analogous, statistical, genesis, comprehensive analysis, and volumetric techniques. Due to the low degree of shale oil exploration in China, the analogous method is unsuitable [20] because of the lack of connection with the actual geological conditions of oil and gas fields. The statistical approach is only suitable for evaluating and predicting oil and gas reservoirs or basin areas in the middle and late stages of exploration and development [21]. The genetic method cannot obtain the residual hydrocarbon occurrence mechanism and enrichment pattern and has parameter dependence in the calculation process, resulting in the accuracy of the calcu-

lation results being constrained [22]. Therefore, the genesis method may not be suitable for resource evaluation of shale reservoirs with low moveability. The integrated analysis method must be based on specific development history information within the exploration study area. This method is challenging to apply in a shale oil and gas exploration area lacking drilling and production data. The volumetric method divides the rocks into unit masses based on a spatial model of hydrocarbon source rocks for the target area of shale oil exploration. The oil content per unit mass is then used to quantitatively evaluate the shale oil resources of the entire hydrocarbon-bearing region [23].

In this paper, the corrected amount of heavy hydrocarbon was obtained by comparing the difference of $S_2$ before and after oil washing based on the room-temperature samples. The cores were treated combined with liquid nitrogen freezing to prevent light hydrocarbon loss. The correction amount of light hydrocarbon was determined by comparing it with the pyrolysis parameters of the room-temperature samples. Based on the characteristic of triple-division between the total oil content and *TOC*, the shale oil resource grading evaluation criteria in the study area were determined. In addition, the volumetric method was used to calculate the total enriched shale oil resources in the $E_1f_2$ of Gaoyou Sag. This study is helpful for the shale oil exploration and development in Gaoyou Sag, Subei Basin.

## 2. Geology Setting

The Subei Basin is the onshore part of the Subei South Yellow Sea Basin, which is bounded to the north and south by the South Jiangsu Uplift and the Lusu Ancient Land, respectively, west to the Tanlu Fault and east to the South Yellow Sea Basin, with an area of $3.8 \times 10^4$ km² (Figure 1) [24]. The Subei Basin is divided into two depressions in the north and south by the Jianhu Uplift, namely the Dongtai and Yanfu Depressions. The Dongtai Depression is divided into six sags (Jinhu, Linze, Gaoyou, Baiju, Qintong, and Haian) and eight salients (Lintangqiao, Liubao, Tuoduo, Wubao, Yuhua, Xiaohai, Taizhou, and Liangduo). The Gaoyou Sag is located in the central part of the Dongtai Depression and the Cenozoic strata, such as the Taizhou ($K_2t$), Funing ($E_1f$), and Dainan ($E_2d$) Formations, are deposited from the bottom up. The shale in $E_1f_2$ is the most developed and the primary exploration area for shale oil in the Subei Basin [25].

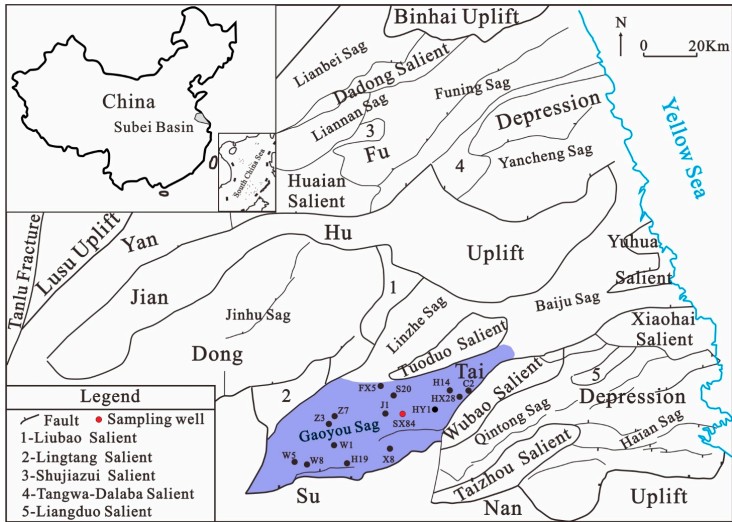

**Figure 1.** Geological setting of the study area [26].

## 3. Methodology

### 3.1. Samples and Experiments

In this study, a total of 119 shale samples were collected from Well SX84 in the Gaoyou Sag. Additionally, 92 samples were collected for the liquefrozen Rock-Eval measurements, which means that the sample pretreatments were conducted at the temperature of liquid nitrogen—at about −196 °C. The *TOC* and Rock-Eval tests were carried out using the

Leco-SC 230 and Rock-Eval VI instruments. Furthermore, the Rock-Eval experiments were performed on the as-received and oil-washed shale samples. This study adopted the mixed solvent of dichloromethane and acetone (3:1 in volume) to eliminate the residual oil in shales under the temperature of 80 °C and the pressure of 0.2 MPa. These samples were then dried in a vacuum oven at 110 °C for 24 h.

*3.2. Correction of Light and Heavy Hydrocarbon Loss for $S_1$*

$S_2$ represents the generated hydrocarbons from kerogen when the temperature gradually rises from 300 °C to 600 °C during rock pyrolysis. However, some heavy hydrocarbons already present in the shale have boiling points much higher than 300 °C. Moreover, the adsorption and swelling of kerogen cause it to be impossible for the residual hydrocarbons to be completely volatilized before 300 °C, corresponding to this residual hydrocarbon being detected as $S_2$. If there is no heavy hydrocarbon loss from the shale, $S_2$ (from room-temperature tests) the values are nearly equal to the $S_{2L}$ (from liquefrozen tests). If the hydrocarbon in the room-temperature sample after being oil-washed is basically removed, the hydrocarbon in $S_{2'}$ is all from kerogen pyrolysis. Therefore, the difference between $S_2$ and $S_{2'}$ is the corrected amount of heavy hydrocarbon for $S_1$. The corrected amount of heavy hydrocarbon can be written as follows:

$$\Delta S_2 = k_2 \cdot S_2, \tag{1}$$

where $k_2$ is the heavy hydrocarbon correction index.

$S_1$ refers to the volatilized hydrocarbons during the rock pyrolysis analysis when temperatures are less than 300 °C, which are $C_{7-33}$ hydrocarbons. However, shale light hydrocarbons ($C_{6-13}$) have primarily dissipated due to the long placement time. The liquefrozen technology is the most effective and straightforward method to correct light hydrocarbons [26]. Since the shale samples were cored on site and then processed at liquid nitrogen temperatures, the light hydrocarbon losses are negligible (the temperature of liquid nitrogen is about −196 °C and the melting point of methane is about −182.6 °C). Subsequently, the $S_{1L}$ was measured from the liquefrozen shale. The light hydrocarbon correction equation can be obtained by establishing a linear relationship between $S_1$ and $S_{1L}$. Therefore, the $S_1$ light hydrocarbon correction model can be determined as follows:

$$S_{1L} = k_1 \cdot S_1, \tag{2}$$

where $k_1$ is the light hydrocarbon correction factor. Then, the total shale oil content ($S_{1O}$) can be obtained.

$$S_{1O} = k_1 \cdot S_1 + k_2 \cdot S_2. \tag{3}$$

*3.3. ΔlogR Method*

The logging evaluation method has the advantages of a shorter cycle, higher resolution, and more visual and continuous characteristics [27]. The ΔlogR method was widely used to estimate organic matter (*TOC*) or oil ($S_1$ or "A") contents in shales based on the acoustic interval transit time (AC) and resistivity (Rt) logging curves. However, using a fixed empirical formula or coefficient to estimate the *TOC* of source rocks is not objective because the properties and well-logging response characteristics of source rocks in different areas are obviously different. Therefore, these methods have been improved in three aspects: the value of the superposition coefficient, the determination of the baseline value, and the control of the influence of maturity. The improved superposition coefficient is determined based on the log data in the study area and the baseline value is automatically identified. [28,29]. The model equation is as follows:

$$\Delta \log R = \lg R + \lg(R_{\max}/R_{\min})/(\Delta t_{\max} - \Delta t_{\min}) \cdot (\Delta t - \Delta t_{\max}) - \lg R_{\min} \tag{4}$$

$$TOC = a \cdot \Delta \log R + b, \tag{5}$$

where $R$ is the value of the resistivity curve, $\Omega\cdot$m; $\Delta t$ is the value of acoustic time difference, $\mu$s/ft; $R_{min/max}$ is the maximum/minimum value of resistivity, $\Omega\cdot$m; $\Delta t_{min/max}$ is the maximum/minimum value of acoustic time difference, $\mu$s/ft; and $a$ and $b$ are the coefficients in the fitted equation, corresponding to the maturity parameter and background value of *TOC* content, respectively. The organic matter of the $E_1f_2$ in the study area has high heterogeneity. Therefore, the improved $\Delta$log$R$ method was used in this study to estimate the *TOC*, $S_1$, and $S_2$ of the shale.

## 4. Results and Discussion

### 4.1. Shale Organic Geochemical Characteristics

The *TOC* contents of the selected shales range from 0.19% to 3.4%, with a mean of 1.02%, mainly in the range of 0–2% (Figure 2). The $S_1$ is between 0.03 mg/g and 2.67 mg/g (mean 0.76 mg/g) and is primarily distributed in a range of 0–1.5 mg/g (Figure 3a). The $S_2$ ranges from 0.15 to 4.9 mg/g, with an average of 1.87 mg/g, mainly in the 1–4 mg/g (Figure 3b).

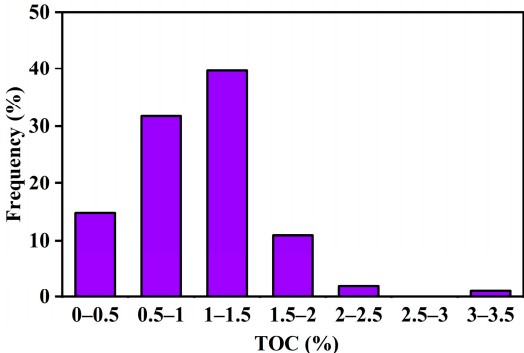

**Figure 2.** Distribution of *TOC* content.

The $S_{1'}$, obtained from the oil-washing samples, range from 0.08 mg/g to 0.19 mg/g, with an average of 0.13 mg/g, which is significantly lower than $S_1$ (Figure 3a,e). The results mean that the residual oil was removed by the oil washing. The $S_{2'}$ (related to oil-washing shales) varies from 0.24 mg/g to 0.81 mg/g, with an average of 0.59 mg/g, which is lower than $S_2$ (Figure 3b,f), implying that there is a part of heavy hydrocarbons that could not be detected in $S_1$ but calculated to $S_2$. The $S_{1L}$ associated with liquefrozen shale varies from 0.05 mg/g to 4.64 mg/g (mean 1.22 mg/g) and is primarily distributed in a range of 0–3 mg/g (Figure 3c). $S_{1L}$ is greater than $S_1$, indicating the light–medium hydrocarbon loss in the room-temperature sample. The corresponding $S_{2L}$ is between 0.21 and 5.24 mg/g, mainly in the 1–4 mg/g range, with an average of 2.23 mg/g (Figure 3d).

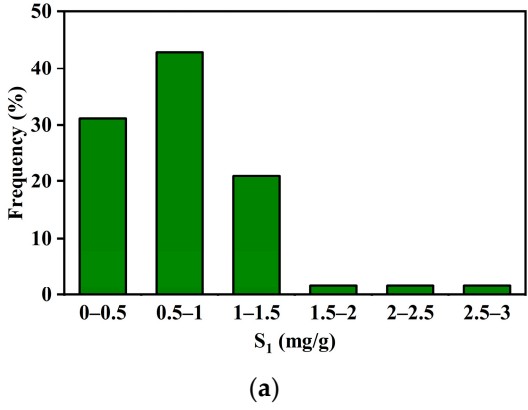

(a)

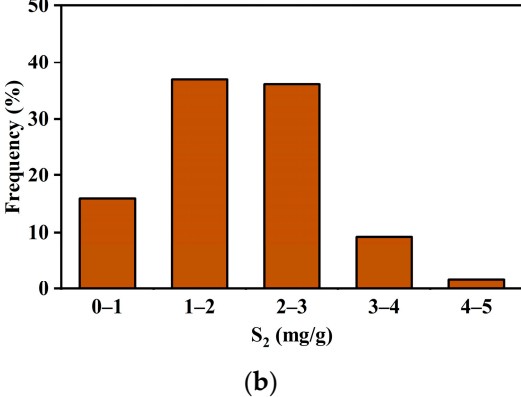

(b)

**Figure 3.** *Cont.*

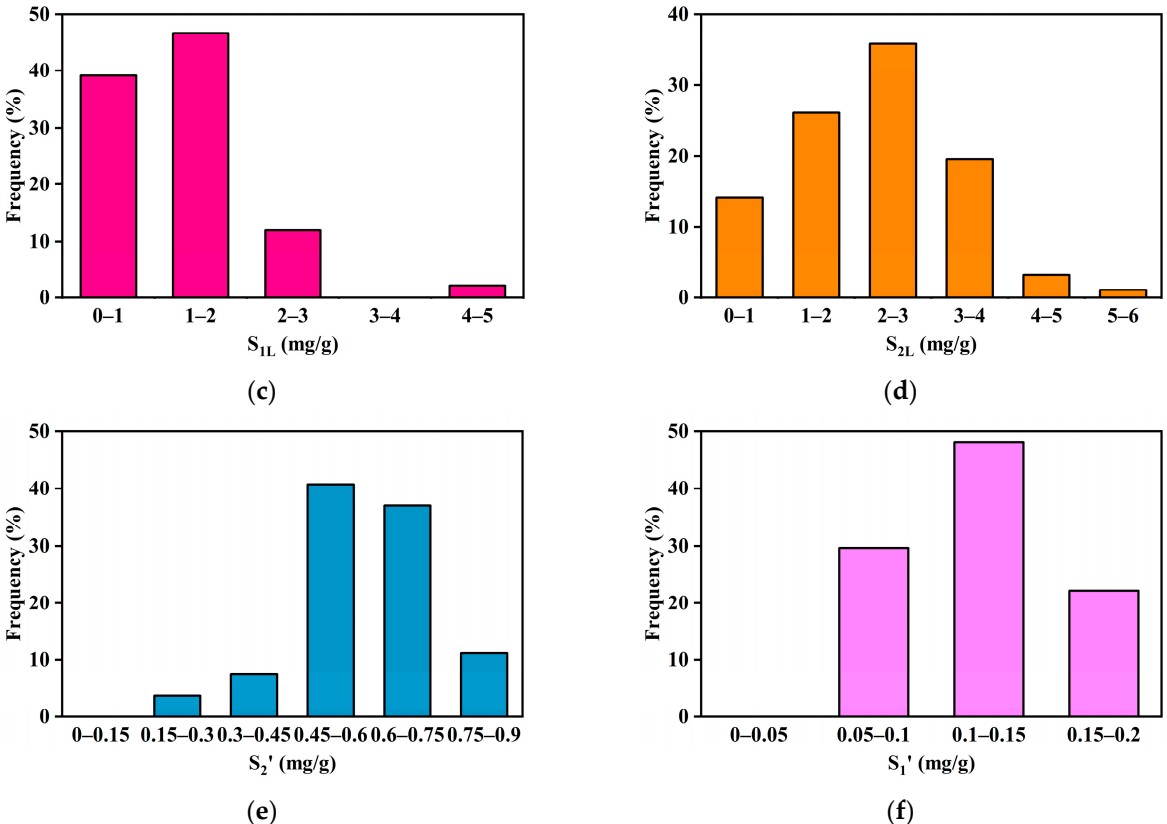

**Figure 3.** Rock-Eval parameters obtained from the liquefrozen, as-received, and oil-washing shales. (**a**) Distribution of $S_1$ content. (**b**) Distribution of $S_2$ content. (**c**) Distribution of $S_{1L}$ content. (**d**) Distribution of $S_{2L}$ content. (**e**) Distribution of $S_{2'}$ content. (**f**) Distribution of $S_{1'}$ content.

### 4.2. Correction Model of Heavy and Light Hydrocarbon Loss for $S_1$

As illustrated in Figure 4a, values of $S_{1'}$ are commonly lower than those of $S_1$, indicating that the residual oil was almost completely removed. However, $S_2$ agrees well with the $S_{2L}$, with a large correlation coefficient ($R^2 = 0.82$) (Figure 4b), implying that almost no heavy hydrocarbons were lost. Therefore, the room-temperature shales can be used to determine the heavy hydrocarbon correction model. The difference ($\Delta S_2$) between $S_2$ and $S_{2'}$ correlates well with $S_2$ with a slope of 0.69 ($R^2 = 0.9$), as exhibited in Figure 5a. Thus, the $S_1$ heavy hydrocarbon correction amount can be determined as about $0.69 \cdot S_2$.

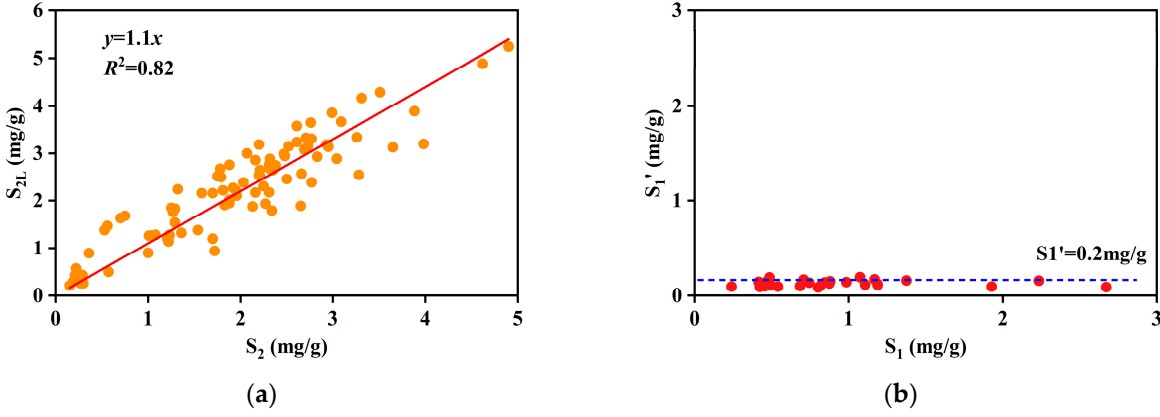

**Figure 4.** Relationships between room-temperature and liquefrozen shales values of $S_2$ (**a**) and relationships between before and after oil washing values of $S_1$ (**b**).

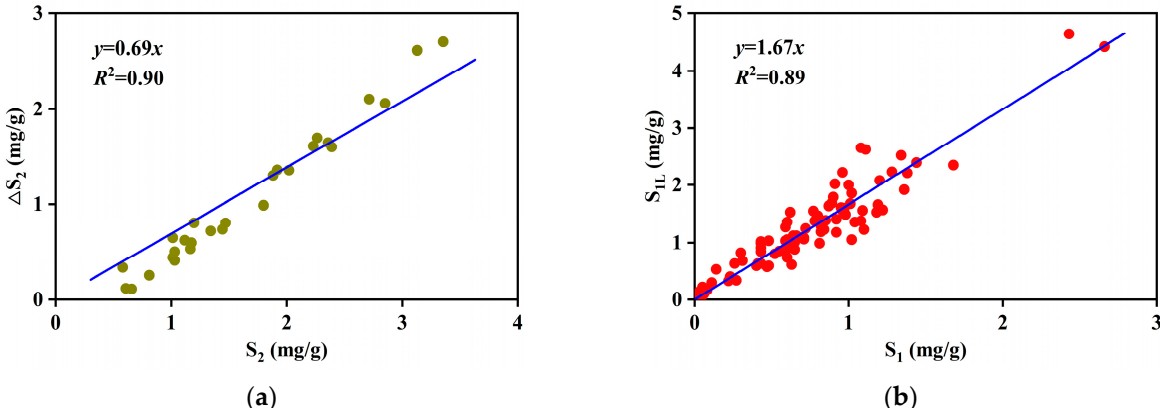

**Figure 5.** Heavy (**a**) and light (**b**) hydrocarbon correction models.

Figure 5b shows that $S_{1L}$ obtained from the liquefrozen shales correlates linearly with $S_1$ received from the room-temperature shales, with a slope of 1.67. Thus, the shale $S_1$ light hydrocarbon correction factor is 1.67 and the light hydrocarbon compensation is $0.67 \cdot S_1$. Additionally, the total shale oil content ($S_{1O}$) is:

$$S_{1O} = 1.67 \cdot S_1 + 0.69 \cdot S_2. \tag{6}$$

### 4.3. Classification of Shale Oil Resource

According to the heavy and light hydrocarbon correction models, the total oil contents in shales were determined. The total shale oil contents range from 0.15 mg/g to 7.63 mg/g (mean 2.55 mg/g), which is about 3.2 times that of tested $S_1$.

As displayed in Figure 6, the shale oil resource classification can be established according to the "trichotomy" between *TOC* and $S_{1O}$. When the *TOC* is low (<0.6%), $S_{1O}$ shows a lower value (<1.8 mg/g), corresponding to the ineffective resource. When the *TOC* increases from 0.6% to 1.5%, $S_{1O}$ gradually increases, corresponding to the low-efficiency resources. However, if the *TOC* is larger than 1.5%, $S_{1O}$ is characterized by a high value corresponding to the enriched resource. Therefore, the shale oil resource classification of the $E_1f_2$ in the Gaoyou Sag was evaluated as follows: enriched resource ($TOC \geq 1.5\%$, $S_{1O} \geq 4.0$ mg/g), low-efficiency resource ($0.6\% \leq TOC < 1.5\%$, 1.8 mg/g $\leq S_{1O} < 4.0$ mg/g), and ineffective resource ($TOC < 0.6\%$, $S_{1O} < 1.8$ mg/g) (Figure 6).

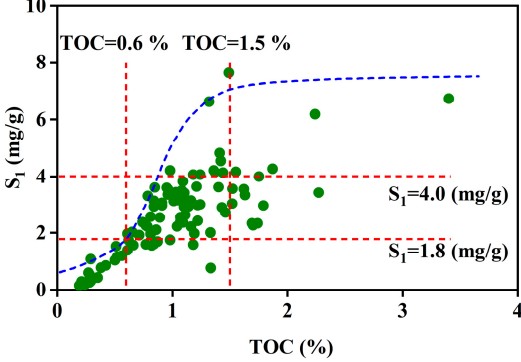

**Figure 6.** Relationship between *TOC* and $S_{1O}$.

According to the resource classification criteria established above, the main type of shale oil in the Gaoyou Sag is inefficient resources.

### 4.4. Estimation of Shale Oil Resource

4.4.1. Calculations of *TOC*, $S_1$, and $S_2$ by $\Delta \log R$

The improved $\Delta \log R$ model was used to estimate the *TOC*, $S_1$, and $S_2$ of the $E_1 f_2$ and the estimated models were determined based on the tested values from the Well SX84 located in the Gaoyou Sag. The reliability of the logging data of the shale section with a thickness of less than 1 m was carefully studied. As exhibited in Figure 7a, the *TOC* shows a significant positive correlation with amplitude difference ($R^2 = 0.85$). The *TOC* can be calculated as follows:

$$TOC = 0.62 \cdot \Delta \log R - 0.27. \tag{7}$$

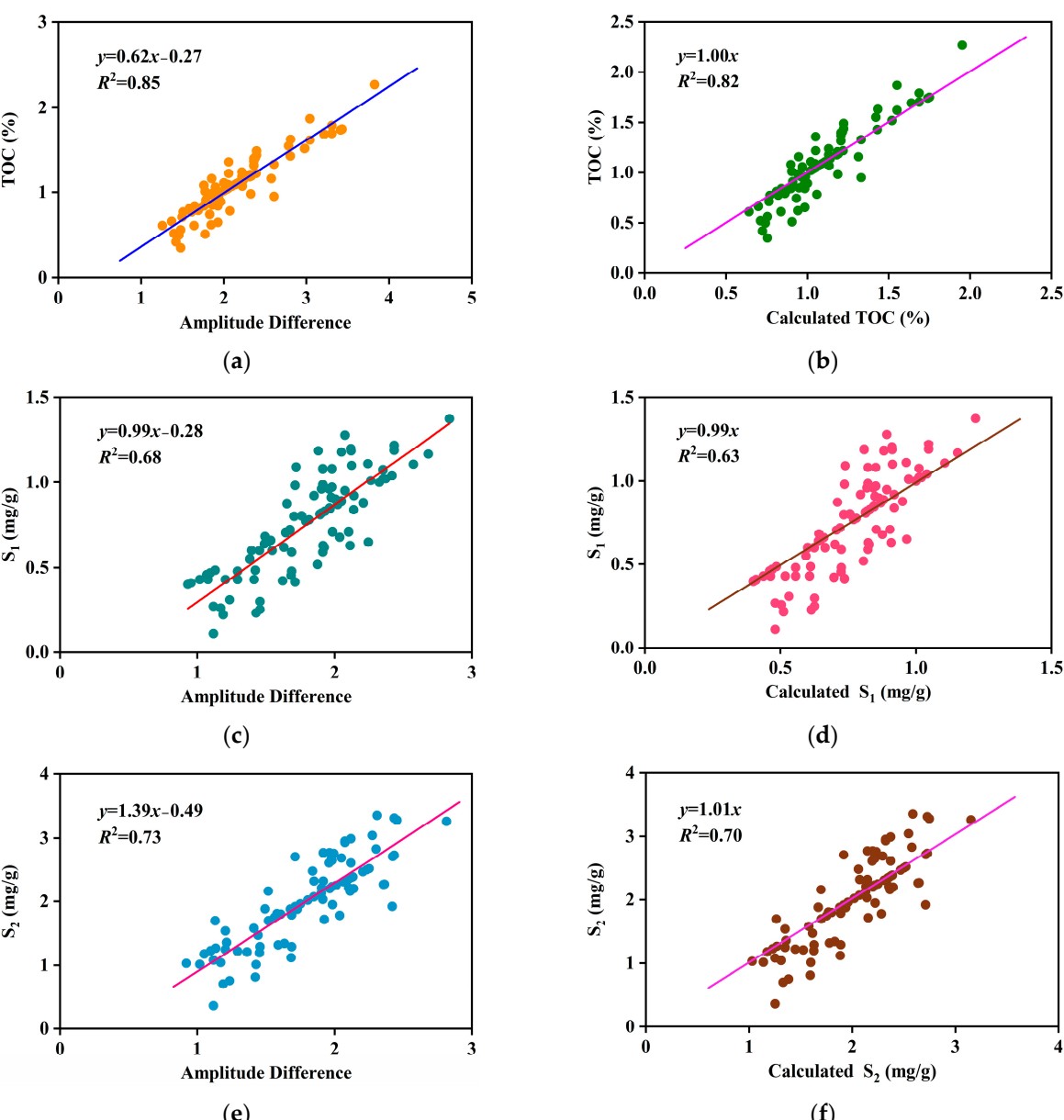

**Figure 7.** Relationships between calculated and measured *TOC*, $S_1$, and $S_2$ (**b,d,f**). Relationship between amplitude difference and *TOC*, $S_1$, and $S_2$ (**a,c,e**).

The estimated *TOC* agrees well with the tested *TOC* ($R^2 = 0.82$), indicating that the calculation values are reliable. The estimated models of $S_1$ and $S_2$ were obtained using the same method. Additionally, the calculated $S_1$ and $S_2$ values show good positive correlations with magnitude differences. Moreover, the calculated $S_1$ values correspond

well with tested $S_1$ and so do the calculated and tested $S_2$, as displayed in Figure 7d,f. The $S_1$ and $S_2$ estimated models are as follows.

$$S_1 = 0.99 \cdot \Delta logR - 0.28. \tag{8}$$

$$S_2 = 1.39 \cdot \Delta logR - 0.49. \tag{9}$$

The established models were applied to estimate the *TOC*, $S_1$, and $S_2$ of the $E_1f_2$ in the Gaoyou Sag. In this study, a total of 17 wells were assessed (Figure 1) and the Wells SX84 and HY1 are shown in Figures 8 and 9. The total shale oil contents of Well HY1 are mainly larger than those of Well SX84. The total shale oil contents of Well HY1 range from 2.32 mg/g to 12.78 mg/g (mean 5.39 mg/g), while the average total shale oil contents in the Well SX84 is 2.46 mg/g (1.12–8.30 mg/g). More resource-rich shale oil occurs in Well HY1.

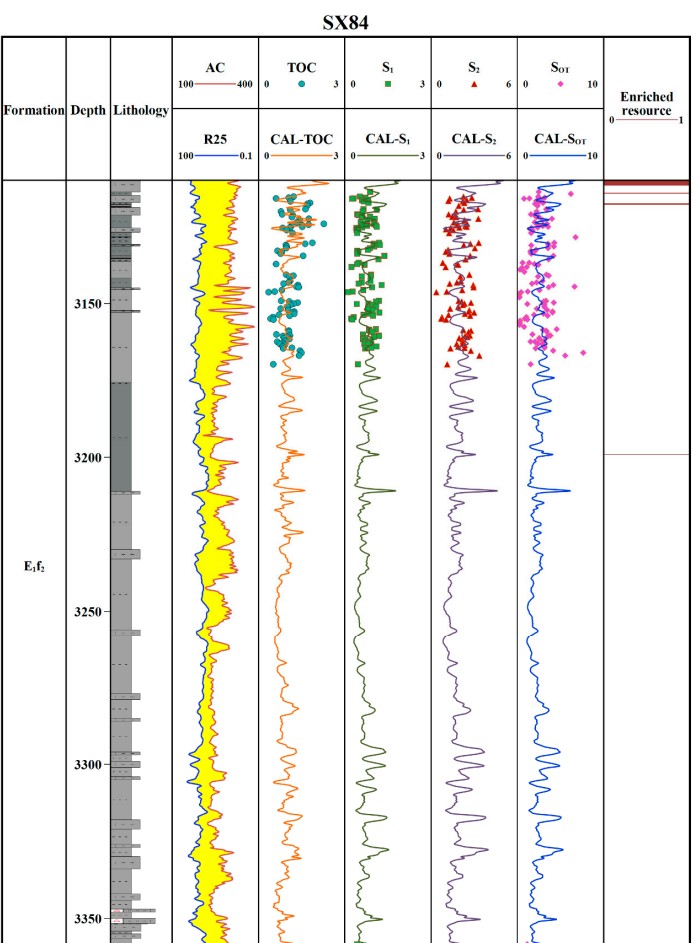

**Figure 8.** Results calculated using the $\Delta logR$ method of Well SX84.

### 4.4.2. Shale Oil Resource in Gaoyou Sag

The total shale oil and enriched resources were calculated using the volumetric method based on the recovered $S_1$ of the $E_1f_2$ in the Gaoyou Sag. The volumetric method of calculating shale oil resources is provided by the following formula:

$$Q = \sum 10^{-1} \times S \times H \times \rho \times S_1, \tag{10}$$

where $Q$ is the single-well shale oil resource, $10^4$ t/km$^2$; $S$ is the shale area, km$^2$; $H$ is the shale thickness, m; and $\rho$ is the shale density, g/cm$^3$. The calculation results show that the total shale oil resources in the study area are about $5.72 \times 10^8$ t, of which the enriched resource is about $1.70 \times 10^8$ t. According to the contour map of shale oil resources in the

Gaoyou Sag (Figure 10), enriched resource areas are mainly in the deep sag, especially near Well HX28.

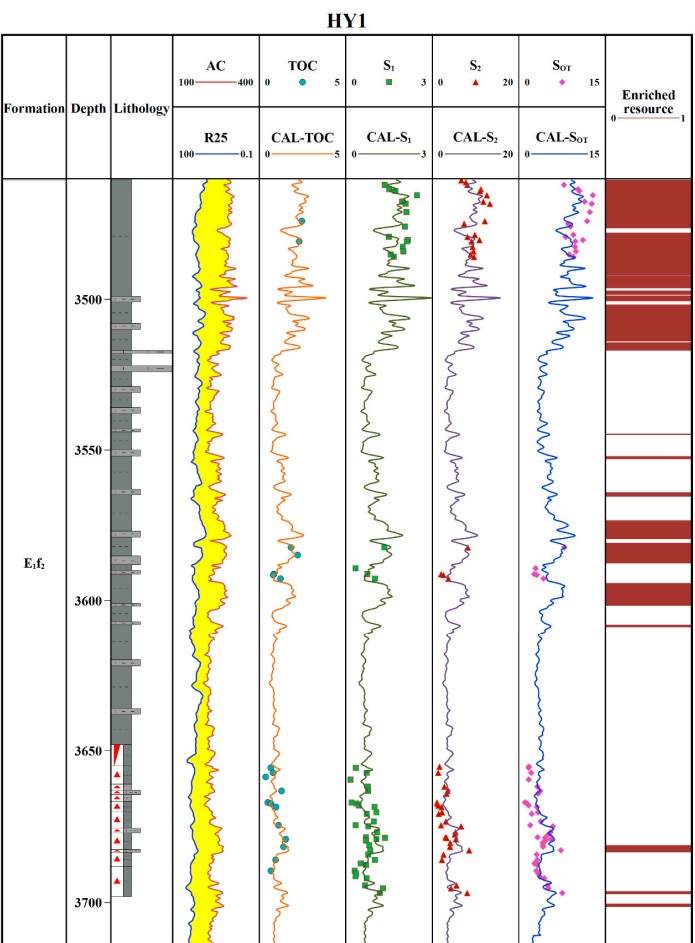

**Figure 9.** Results calculated using the Δlog*R* method of Well HY1. (The red triangles in Lithologic series mean oil immersion).

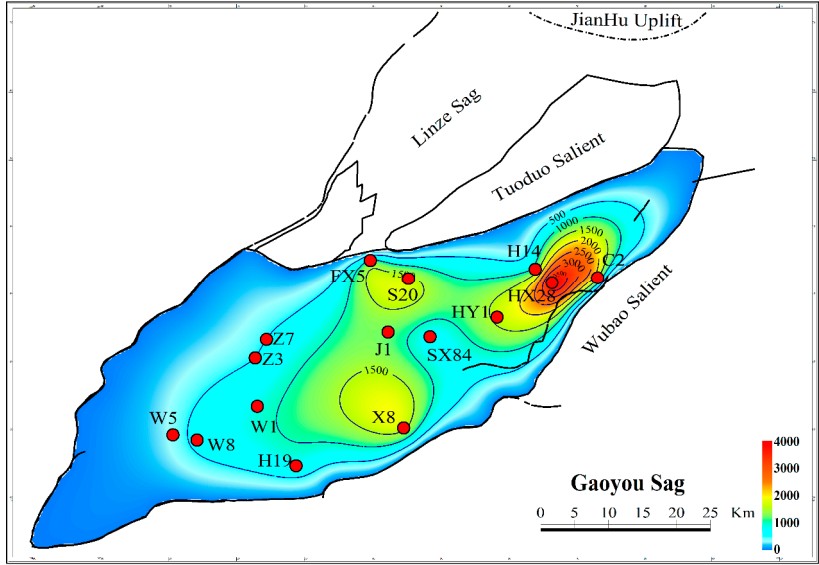

**Figure 10.** Contour map of the oil resource in Gaoyou Sag.

## 5. Conclusions

The light and heavy hydrocarbon correction models of $S_1$ were developed based on the rock pyrolysis of liquefrozen, conventional, and oil-washed shales. The shale $S_1$ heavy hydrocarbon correction content is about $0.69 \cdot S_2$ and the light hydrocarbon compensation is $0.67 \cdot S_1$. The corrected $S_1$ is about 3.2 times more than the conventional shale-tested value.

The shale oil enriched resource classification of the $E_1f_2$ in the Gaoyou Sag was $TOC \geq 1.5\%$ and $S_{1O} \geq 4.0$ mg/g, the low-efficiency resource was $0.6\% \leq TOC < 1.5\%$ and $1.8$ mg/g $\leq S_{1O} < 4.0$ mg/g, and the ineffective resource was $TOC < 0.6\%$ and $S_{1O} < 1.8$ mg/g.

The total shale oil and enriched resources were calculated using the volumetric method. The total shale oil resource in the Gaoyou Sag is about $5.72 \times 10^8$ t, of which the enriched resource is about $1.70 \times 10^8$ t. The enrichment areas are mainly in the deep sag, especially near Well HX28.

**Author Contributions:** Conceptualization, Q.Z. and S.L.; methodology, Q.Z.; software, Q.Z.; validation, Q.Z., P.Z. and H.H.; formal analysis, Q.Z.; investigation, J.W.; resources, J.W.; data curation, Z.L.; writing—original draft preparation, Q.Z.; writing—review and editing, Q.Z.; visualization, Q.Z.; supervision, P.Z.; project administration, S.L.; funding acquisition, S.L. All authors have read and agreed to the published version of the manuscript.

**Funding:** This research was funded by the Natural Science Foundation of Shandong Province (ZR2020QD036). And The APC was funded by S.L.

**Institutional Review Board Statement:** Not applicable.

**Informed Consent Statement:** Informed consent was obtained from all sub-jects involved in the study.

**Data Availability Statement:** Not applicable.

**Conflicts of Interest:** The authors declare no conflict of interest.

## Nomenclature

| | |
|---|---|
| $S_1$ | residual hydrocarbon obtained from room-temperature shale using rock pyrolysis, mg/g; |
| $S_2$ | hydrocarbon from kerogen pyrolysis obtained from room-temperature shale using rock pyrolysis, mg/g; |
| $S_{1L}$ | residual hydrocarbon obtained from liquefrozen shale using rock pyrolysis, mg/g; |
| $S_{2L}$ | hydrocarbon from kerogen pyrolysis obtained from liquefrozen shale using rock pyrolysis, mg/g; |
| $S_{1'}$ | residual hydrocarbon obtained from room-temperature shale after oil-washed using rock pyrolysis, mg/g; |
| $S_{2'}$ | hydrocarbon from kerogen pyrolysis obtained from oil-washed shale using rock pyrolysis, mg/g; |
| $\Delta S_2$ | difference between $S_2$ and $S_2{}'$, mg/g; |
| $S_{1O}$ | total oil, mg/g. |

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
