# Peer review of "Correction of Light and Heavy Hydrocarbons and Their Application in a Shale Oil Reservoir in Gaoyou Sag, Subei Basin—A Case Study from Well SX84"

_processes, doi:10.3390/pr11020572_

Round 1
Reviewer 1 Report
The work is very interesting. Many (119) samples of shale rocks have been analyzed, which causes great respect. The authors developed the light and heavy hydrocarbon correction models of S1 and S2 parameters and calculated by the volumetric method the total shale oil and enriched resources of the E1f2 in the Gaoyou Sag.
There are some wishes:
Most likely, the name "--" is a typo. First impression from the name: it's not entirely clear what "Correction of light and heavy hydrocarbons" means. It seems to me that it is worth revising the name in accordance with the purpose of the work (why this correction is necessary).
Sincerely, Reviewer
Reviewer 2 Report
In this work, the authors used multiple methods to correct the S1 values and then establish a log-based methods to predict the S1, S2 and TOC values of shale. Overall, this method is well presented and provides some useful information for evaluating the shale oil content. Below are my comments.
1. In the introduction, the authors should introduce the recent advances in using well logs to predict the TOC, S1 and S2.
2. A lithology profile should be provided along with the structural map in Fig. 1.
3. The authors should give more detailed information on how the Eq (1) and Eq (2) are established.
4. Caption of Fig. 2. It should be distribution of TOC content.
5. The nomenclatures and its physical meaning are used randomly, and the authors should clearly introduce the meanings of these nomenclatures in the section of methods. For example, in section 3.2, what is S1? In section 4.1, S1’, S2’ and S2L comes out. I strongly advised the authors to introduce their physical meanings beforehand.
6. In establishing the mathematical correlation between S2 and dS2, the authors only used 20 data points. So where are the left 70+ data points? Similar problems can also be found in Fig. 5b and Fig. 4. So I highly doubt the reliability of the Eq. (6).
7. In section 4.3, a clear conclusion should be given regarding the type of shale oil in Gaoyou Sag. For example, the main type of shale oil is low efficiency resource.
8. The accuracy of LogR method in estimating the TOC content of shale is highly dependent on the parameters of baseline values. In the past, many studies have reported the unreliability of logR or improved logR method in estimating the TOC content. In this work, the calculated TOC has a great correlation with measured TOC, with R2=0.82. It is very surprised. The authors should clearly introduce that how the parameters of logR method, such as baseline values, are determined and what the values are.
